

# Correlation between hematological indicators in acclimatized high-altitude individuals and acute mountain sickness

Zhicai Li[1,2], Jun Xiao[2], Cuiying Li[1,2], Xiaowei Li[2] and Daoju Ren[1,2]

[1] Air Force Clinical College; The Fifth School of Clinical Medicine, Anhui Medical University, Hefei, China
[2] Department of Blood Transfusion, Air Force Medical Center, Air Force Medical University, Beijing, China

Corresponding authors
Jun Xiao, ammsxj@fmmu.edu.cn
Cuiying Li, lcy2013@fmmu.edu.cn

## ABSTRACT

**Background:** The impact of acute mountain sickness (AMS) on individuals ascending to plateaus, soon after exposure to high altitudes, is well-documented. However, the specific relationship between AMS and alterations in blood parameters remains unclear.

**Methods:** A total of 40 healthy volunteers were recruited. Following their arrival at an altitude of 3,300 m, an AMS questionnaire survey was administered 48 h later. Based on the AMS scores obtained, participants were categorized into three groups: non-AMS, mild AMS, and moderate/severe AMS (encompassing both moderate and severe cases). Blood routine tests were performed on all groups at 3-, 7-, and 30-days post-arrival at the plateau, with blood oxygen saturation tests conducted at 3 and 30 days after rapidly entering the plateau.

**Results:** In the current investigation, a total of 40 participants were stratified into non-AMS ($n = 24$), mild-AMS ($n = 8$), and moderate/severe-AMS ($n = 8$) cohorts subsequent to rapid ascension to an altitude of 3,300 m. The incidence of AMS in this study was 40%. Noteworthy elevations in red blood cells (RBC), hemoglobin (Hb), and hematocrit (HCT) levels were noted at the 3-day mark post-ascent across all delineated groups. By the 7th day, the moderate/severe-AMS cohort displayed sustained increments in Hb and HCT levels, whereas solely HCT levels rose in the mild-AMS and non-AMS cohorts. Upon reaching the 30-day milestone, the moderate/severe-AMS group demonstrated a reduction in RBC, Hb, and HCT levels, while only HCT levels decreased in the mild-AMS and non-AMS groups. Furthermore, it was observed that all groups exhibited notable reductions in oxygen saturation ($SpO_2$) at 3 days post-ascent, followed by a partial recovery at 30 days, albeit remaining below baseline levels. The correlation analysis results indicated that RBC, Hb, and HCT exhibited a positive correlation with the severity of AMS after a 7-day acclimatization period at high altitude. Conversely, $SpO_2$ demonstrated a negative correlation with the severity of AMS following the same duration at high altitude. The findings of the study suggest a strong association between alterations in RBC, Hb, and HCT levels and AMS, particularly among individuals in the moderate/severe-AMS category who displayed more significant fluctuations in these parameters.

**Conclusion:** Individuals suffering from moderate to severe AMS demonstrated increased levels of RBC, Hb, and HCT, as well as reduced $SpO_2$, indicating a greater

need for oxygen adaptation to high-altitude hypoxia. These findings emphasize the physiological adjustments to high altitudes and their potential implications for the treatment of AMS.

## INTRODUCTION

The phenomenon of rapid ascent to high altitude (>2,500 m) involves individuals transitioning from lower elevations to higher elevations, or from high altitudes to even greater heights (exceeding 3,500 m), in a condensed timeframe (*Luks, Swenson & Bartsch, 2017*). Following such rapid ascents, the body's adaptation mechanisms are not fully established, leading to challenges in adjusting to the reduced atmospheric pressure and oxygen levels, resulting in symptoms of acute mountain sickness (AMS) such as dizziness, headache, and heightened respiratory rate (*Caravedo et al., 2022*; *Luks, Swenson & Bartsch, 2017*). On the one hand, hypoxia stimulates the sympathetic nervous system, resulting in an increased heart rate and an increase in pulmonary artery pressure to increase the body's ventilation and oxygen supply. On the other hand, it will also trigger the hypoxia induction mechanism, with a large increase in hypoxia-inducible factor (HIF), and an increase in the synthesis of its downstream target genes such as erythropoietin (EPO) and vascular endothelial growth factor (VEGF), which induces erythrocyte and angiogenesis, and increases tissue oxygen supply to adapt to the hypoxic environment (*Li, Zhang & Zhang, 2018*). At 6–12 h after the rapid advance to the plateau, there are certain individual differences in the body's response to hypoxia. In terms of altitude sickness, symptoms may be mild, with significant AMS, and severe may occur as pulmonary edema and cerebral edema.

Some studies have suggested that hematological parameters such as reticulocyte and neutrophil counts at sea level may be associated with high altitude headache (HAH), a cardinal symptom of AMS (*Wang et al., 2018*). This implies that individuals with certain baseline hematological profiles might be more susceptible to AMS. Additionally, hematological parameters like white blood cell (WBC) count have been linked to inflammation, which is believed to play a role in the etiology of AMS (*Yan et al., 2024*). Higher WBC counts have been associated with diabetes, and since inflammation is a common feature in diabetic patients, this could indirectly suggest a link between hematological parameters and AMS risk (*Yan et al., 2024*). Hematological parameters, particularly those related to oxygen-carrying capacity like hemoglobin, are crucial in high-altitude environments where oxygen levels are low. Reduced oxygen saturation (SpO$_2$) is a risk factor for HAH, which is part of the AMS spectrum (*Wang et al., 2018*). This suggests that hematological parameters influencing oxygen delivery could be associated with AMS.

In terms of hemoglobin, there may be significant changes or minor changes. Recent studies have shown that the low level of hemoglobin concentration in the Tibetan

population living in the plateau is related to the polymorphism of HIF-2α, and the population carrying polymorphic loci shows smaller changes in hemoglobin concentration after entering the plateau (*Li et al., 2019*). Reexposure to altitude after hypoxic training with AMS reduces the symptoms of AMS (*Burtscher, Hefti & Hefti, 2021*).

The classification of AMS into mild, moderate, and severe categories reveals statistically significant variations in hematological parameters among individuals with differing levels of AMS severity (*Wang et al., 2018*). Hemoglobin and oxygen saturation are important hematological indicators of altitude adaptation, while alterations in lung function are also associated with the onset of AMS. Prior research has primarily concentrated on general hematological alterations and pulmonary function parameters with AMS following rapid ascent (*Li et al., 2018*; *Reiser et al., 2024*; *Seiler et al., 2023*), but there remains a gap in the literature regarding specific investigations into hematological changes among individuals experiencing varying degrees of AMS.

Therefore, this study aims to examine hematological indicators in individuals experiencing diverse high-altitude responses after a rapid ascent to elevated altitudes. It further seeks to investigate the correlation between these hematological indicators and AMS by conducting blood routine tests and measuring blood oxygen saturation in a cohort of 40 individuals at 3, 7, and 30 days post-ascent.

# MATERIALS AND METHODS

## Population

Forty healthy adults (18–36 years) from Beijing Sport University, with a mean age of 23.9 ± 4.4 years, were selected to participate in the study due to their intention to physical training to high altitudes (≥2,500 m). Before entering the plateau, they often live at an average altitude of 43.5 m above sea level, an average temperature 17.3 °C and average humidity 75%. Exclusion criteria encompassed a history of high-altitude hypoxia exposure, a body mass index (BMI) exceeding 25, hematological disorders, kidney and bone disease, professional athlete, primary headaches, heart failure, respiratory system disorders, and recent illnesses such as colds. The study included participants of both genders. Women were measured during the early follicular phase of their menstrual cycle to standardize the hormonal influence on hematological parameters. The research involving human subjects received approval from the Institutional Review Board of the Air Force Medical Center (NO. 2022-05-YJ07). The studies were conducted according to the guidelines of the Declaration of Helsinki and its later amendments. All participants provided signed written informed consent.

## AMS survey

After entering the plateau environment, strictly control the intake of drugs and foods with nervous system effects, such as alcohol, coffee, acetaminophen. In conducting this study, a survey instrument pertaining to high-altitude reactivity was constructed utilizing the Lake Louise Acute Mountain Sickness Score (*Roach et al., 2018*), which removes the part of sleep disturbance in acute altitude sickness and recommends that altitude sickness symptoms be recorded 6 h after a rapid altitude entry. Participants completed the questionnaire in a

self-report format within 48 h of ascending rapidly to an altitude of 3,300 m. The questionnaire encompassed the following domains: (1) Demographic Information: Gender, age, ethnicity; and (2) symptoms of AMS: presence of symptoms including headache, dyspnea, chest discomfort, nausea, vomiting, sleep disturbances, and tinnitus.

## Grouping of the study

The study participants were categorized into distinct groups according to the severity of AMS. These groupings included: (1) the non-AMS group, consisting of individuals who exhibited no symptoms of altitude sickness and obtained a total score between 1 and 4; (2) the mild-AMS group, comprising individuals with mild AMS symptoms such as mild headache, 1–2 episodes of vomiting per day, or a total score ranging from 5 to 10; and (3) the moderate/severe-AMS group, consisting of individuals with moderate to severe AMS symptoms. The inclusion criteria for this group were severe headache, or three or more episodes of vomiting per day, or a total score of 11 or higher (*Talks et al., 2022*).

## Treatment of AMS

Before ascending to high altitudes, it is not recommended to use prophylactic measures such as Rhodiola or other medications for AMS. Upon arrival at high altitudes, individuals without AMS or those with mild symptoms may not require any intervention. Oxygen therapy may be considered during periods of exacerbation. For individuals experiencing moderate to severe AMS, continuous oxygen therapy or a combination of diuretics and corticosteroids may be administered based on symptom severity. In cases of worsening AMS, hospitalization for treatment is advised.

## Hematological analysis and oxygen saturation assessment

The samples were collected indoors every times, with a temperature of about 23 °C, a humidity of 30%, an altitude of 3,300 m, and an atmospheric pressure of 65.2 kpa. Drinking and fasting should be avoided for 12 h before sampling. Blood samples were collected in EDTA tubes and stored at 4 °C until analysis. To ensure consistency, samples were analyzed within 2 h of collection. Hematological analysis utilizes EDTA-$K_2$ anticoagulant and automated hematology analyzers (XN-530X; Sysmex Corporation, Hyogo, Japan) for blood routine testing. Oxygen saturation assessment is performed with a finger pulse oximeter (CMS50N; CONTEC Company, Osaka, Japan), which is meticulously maintained and calibrated by a specialized professional. All hematological analyses were performed in a climate-controlled laboratory with a stable temperature (16–24 °C) and humidity (<80%). Hematological analyzers were calibrated daily using standard controls, and all assays were performed according to the manufacturer's instructions. Internal controls were included with each batch to monitor assay performance. Prior to assessment, subjects are instructed to be in a resting state. Each subject's measurement is repeated three times and the average value is calculated. These assessments are carried out at specific time intervals, including baseline, 3 and 7 days after rapid ascent, and 30 days post-event.

**Table 1 The general characteristics of the volunteers.**

| Characteristics | | AMS | | | F/H | P |
|---|---|---|---|---|---|---|
| | | None (*n* = 24) | Mild (*n* = 8) | Moderate/Severe (*n* = 8) | | |
| Age (years) | | | | | 6.105 | 0.047 |
| | ≤20 | 8 | 0 | 3 | | |
| | 21–30 | 16 | 6 | 4 | | |
| | ≥31 | 0 | 2 | 1 | | |
| Race | | | | | – | 0.198 |
| | Han | 18 | 8 | 8 | | |
| | others | 6 | 0 | 0 | | |
| Gender | | | | | – | 0.917 |
| | Male | 14 | 6 | 4 | | |
| | Female | 10 | 4 | 4 | | |
| Height | | 171.72 ± 4.99 | 171.09 ± 4.52 | 175.06 ± 4.04 | 1.798 | 0.180 |
| Weight | | 63.58 ± 5.34 | 64.20 ± 5.47 | 66.01 ± 5.40 | 0.613 | 0.547 |
| BMI | | 21.58 ± 1.81 | 21.93 ± 1.55 | 21.56 ± 1.96 | 0.126 | 0.882 |

**Note:**
The quantitative data conformed to a normal distribution and are presented as mean ± SD. *F*: the statistics of ANOVA analysis; *H*: the statistics of Kruskal–Wallis test.

## Statistical analysis

Hematological parameters and oxygen saturation variances among non-AMS, mild-AMS, and moderate/severe-AMS groups were assessed utilizing SPSS (version 26.0). Continuous data exhibiting a normal distribution are expressed as mean ± standard deviation ($\bar{x} \pm s$). Independent samples were tested by analysis of variance (ANOVA) for normally distributed variables and repeated measures ANOVA (RM-ANOVA) to test for significant differences in the outcomes over time, with *post-hoc* pairwise comparisons conducted using the least significant difference (LSD) test. Time served as the repeated factor in evaluating the impact of parameters on the outcome variables across different time. The Kruskal–Wallis nonparametric test and Fisher test were used to analyze the distribution differences of different age, race and gender in patients with different degrees of AMS. Spearman correlation analysis was used for correlations. Statistical significance in observed differences was determined at a significance level of $P < 0.05$.

## RESULTS

### Incidence of AMS following rapid ascent to high altitude

Table 1 provides an overview of the basic characteristics of the study participants. This investigation encompassed a total of 40 study participants, among whom 24 individuals (60%) exhibited no symptoms of AMS. Eight participants (20%) experienced mild AMS, while another eight individuals (20%) manifested moderate to severe AMS. No participants received oxygen therapy or medicine therapy. The age distribution varies significantly among individuals with differing degrees of AMS ($P = 0.047$). However, no

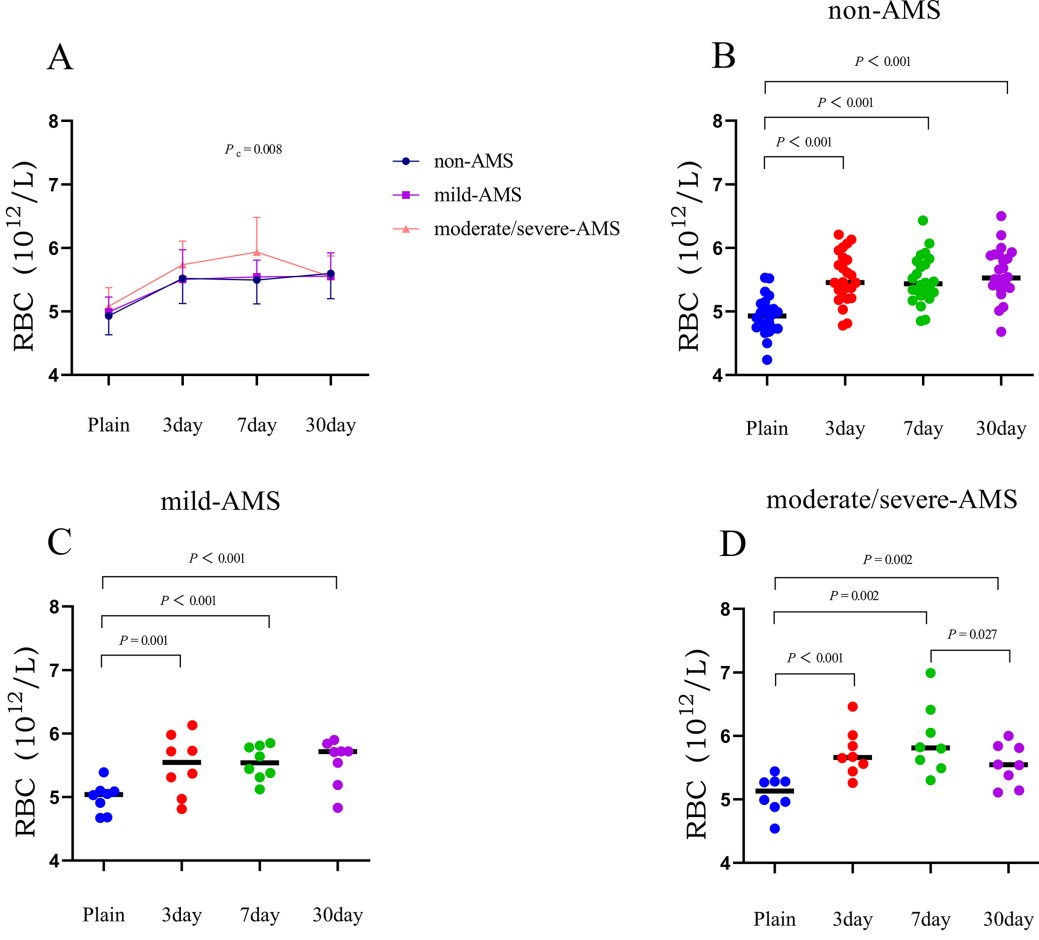

**Figure 1** **The trends in mean changes of RBC following rapid ascent to high altitude.** (A) The trends of RBC variation in different groups before and after entering plateau. The difference of RBC at different time points in non-AMS (B), mild-AMS (C), and moderate/severe-AMS (D).

significant differences were observed in terms of gender, racial distribution, height, weight, and BMI.

## The trends in mean changes of RBC, Hb, and HCT in different group following rapid ascent to high altitude

Within 3 days of acclimating to high altitude, all three groups demonstrated a significant increase in mean values of red blood cell (RBC), hemoglobin (Hb), and hematocrit (HCT) compared to their initial levels. Specifically, the non-AMS group showed increases of $0.6 \times 10^{12}/L$ in RBC, 19.6 g/L in Hb, and 7.3% in HCT compared to baseline. The mild-AMS group had corresponding increases of $0.5 \times 10^{12}/L$ in RBC, 16.4 g/L in Hb, and 6.8% in HCT. The moderate/severe-AMS group experienced increases of $0.7 \times 10^{12}/L$ in RBC, 20.8 g/L in Hb, and 7.8% in HCT ($P < 0.01$; Figs. 1–3, and Table 2).

Between 3 and 7 days following a rapid ascent, individuals categorized in the moderate/severe-AMS group exhibited a continued rise in Hb and HCT levels, whereas those in the

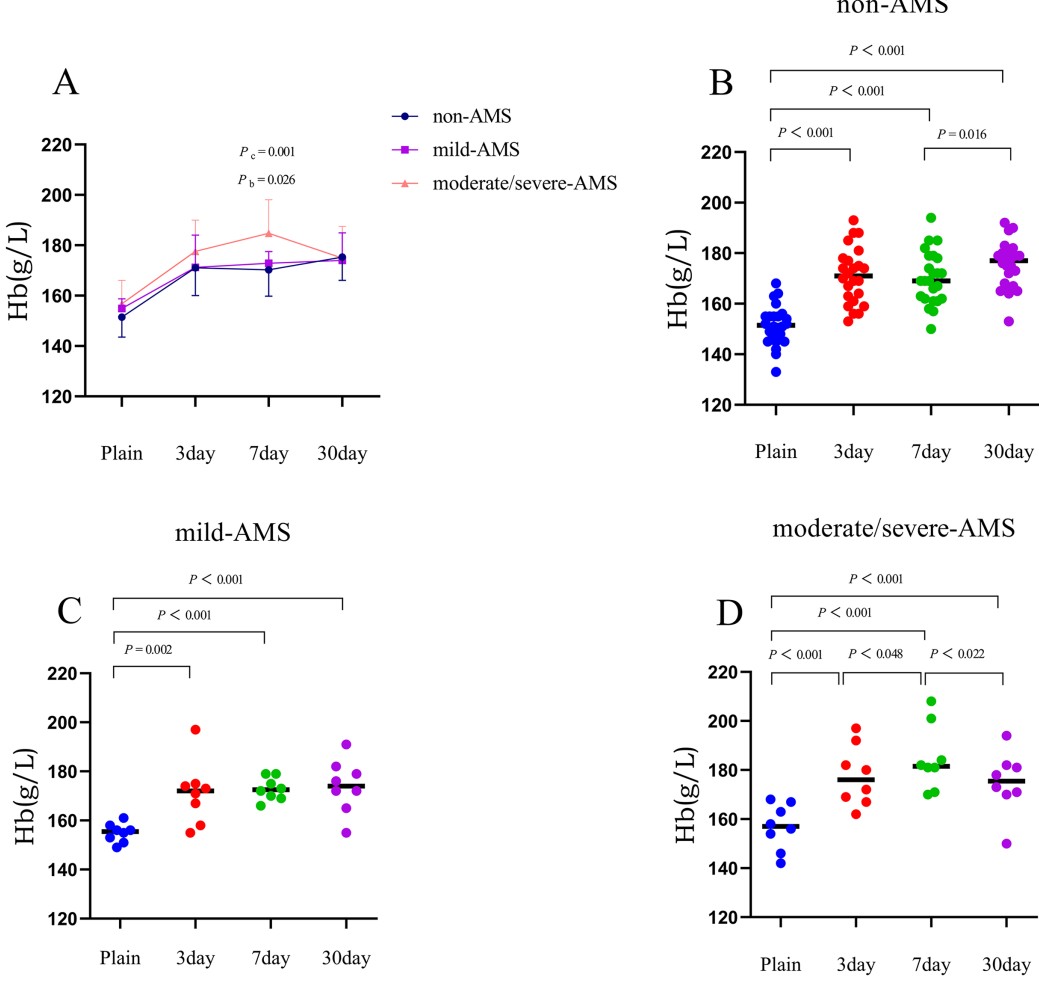

**Figure 2 The trends in mean changes of Hb following rapid ascent to high altitude.** (A) The trends of Hb variation in different groups before and after entering plateau. The difference of Hb at different time points in non-AMS (B), mild-AMS (C), and moderate/severe-AMS (D).

mild-AMS and non-AMS groups only displayed an increase in HCT levels. By the seventh day at high altitude, the non-AMS group demonstrated a 2.2% increase in mean HCT compared to the third day, while the mild-AMS group showed a 3.4% increase. In contrast, the moderate/severe-AMS group experienced a notable increase of 7.1 g/L in Hb and 4.3% in HCT compared to the third day ($P < 0.05$; Figs. 1–3, and Table 2).

Between 7 and 30 days following rapid ascent, individuals in the moderate/severe AMS group experienced a notable reduction in mean RBC, Hb, and HCT, while those in the mild-AMS group exhibited a slight increase in Hb and a decrease in HCT. The non-AMS group only displayed a decrease in HCT. After 30 days at high altitude, the moderate/severe-AMS group demonstrated a decrease in mean RBC, Hb, and HCT by $0.4 \times 10^{12}$/L, 9.9 g/L, and 8.8%, respectively, compared to 7 days. The mild-AMS group saw an increase in mean Hb by 5.1 g/L and a decrease in mean HCT by 5.7%. The non-AMS group experienced a decrease in mean HCT of 4.2% ($P < 0.05$; Figs. 1–3, and Table 2).

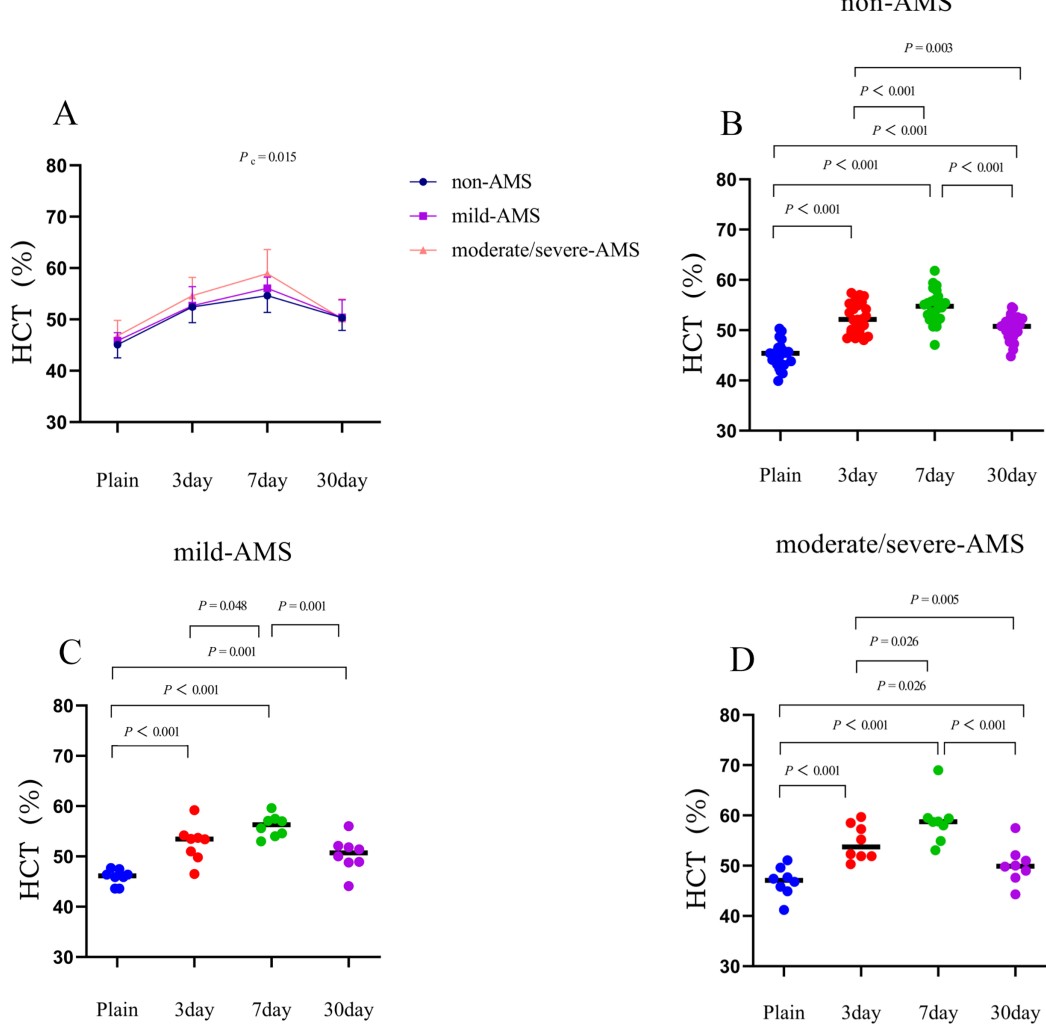

**Figure 3** The trends in mean changes of HCT following rapid ascent to high altitude. (A) The trends of HCT variation in different groups before and after entering plateau. The difference of HCT at different time points in non-AMS (B), mild-AMS (C), and moderate/severe-AMS (D).

## Differences in RBC, Hb, and HCT among the three groups at different time points

At sea level, individuals who did not experience acute mountain sickness (AMS) had average red blood cell (RBC), hemoglobin (Hb), and hematocrit (HCT) values of $4.9 \times 10^{12}$/L, 151.5 g/L, and 45.1%, respectively. Those with mild AMS displayed average values of $5.0 \times 10^{12}$/L, 154.9 g/L, and 45.9%, while individuals with moderate/severe AMS had average values of $5.1 \times 10^{12}$/L, 156.8 g/L, and 46.8%, respectively. The mean values of these parameters among the three groups did not demonstrate any statistically significant differences ($P > 0.05$; Table 3).

On the third day following rapid ascent to high altitude, individuals who did not exhibit AMS displayed mean RBC, Hb, and HCT values of $5.5 \times 10^{12}$/L, 171.0 g/L, and 52.4%, respectively. In contrast, those with mild AMS had mean values of $5.5 \times 10^{12}$/L, 171.3 g/L,

**Table 2 Comparison of mean differences in RBC, Hb, HCT, and SpO$_2$ over different time intervals among individuals experiencing rapid altitude gain.**

| Parameters | Time | Non-AMS (24) | Mild AMS (8) | Moderate/Severe AMS (8) | P value (A) | P value (B) | P value (C) |
|---|---|---|---|---|---|---|---|
| $\Delta$RBC ($10^{12}$/L) | Plain–3 day | 0.60 ± 0.3 | 0.5 ± 0.3 | 0.7 ± 0.3 | 0.000 | 0.001 | 0.000 |
| | 3–7 day | 0 ± 0.3 | 0 ± 0.4 | 0.2 ± 0.3 | >0.05 | >0.05 | >0.05 |
| | 7–30 day | 0.1 ± 0.3 | 0 ± 0.3 | −0.4 ± 0.4 | >0.05 | >0.05 | 0.027 |
| $\Delta$Hb (g/L) | Plain–3 day | 19.6 ± 10.8 | 16.4 ± 9.8 | 20.8 ± 8.6 | 0.000 | 0.002 | 0.000 |
| | 3–7 day | −0.8 ± 7.4 | 1.6 ± 11.5 | 7.1 ± 8.4 | >0.05 | >0.05 | 0.048 |
| | 7–30 day | 5.1 ± 9.3 | 1.1 ± 10.4 | −9.9 ± 9.6 | 0.016 | >0.05 | 0.022 |
| $\Delta$HCT (%) | Plain–3 day | 7.3 ± 3.0 | 6.8 ± 2.5 | 7.8 ± 2.6 | 0.000 | 0.000 | 0.000 |
| | 3–7 day | 2.2 ± 2.5 | 3.4 ± 4.0 | 4.3 ± 4.3 | 0.000 | 0.048 | 0.026 |
| | 7–30 day | −4.2 ± 2.9 | −5.7 ± 3.0 | −8.8 ± 2 | 0.000 | 0.001 | 0.000 |
| $\Delta$SpO$_2$ (%) | Plain–3 day | −8.3 ± 3.4 | −10.5 ± 3.7 | −8.8 ± 2.6 | 0.000 | 0.000 | 0.000 |
| | 3–30 day | 2 ± 4.5 | 4.9 ± 4.1 | 3.0 ± 3.0 | 0.000 | 0.011 | 0.025 |

**Note:**

Intra-group comparisons were conducted for the changes in blood parameters at different time points after rapid ascent to high altitude. The quantitative data conformed to a normal distribution and are presented as mean ± SD. $\Delta$ represents the difference obtained by subtracting the value at a subsequent time point from the value at a preceding time point. P value (A) indicates the P-value for intra-group comparisons in non-AMS group. P value (B) represents the P-value for intra-group comparison in the mild AMS group. P value (C) denotes the P-value for intra-group comparison in the moderate/severe AMS group. AMS, acute mountain sickness.

**Table 3 Statistical analysis of RBC, Hb, HCT, and SpO$_2$ among different groups following rapid ascent to high altitudes.**

| Parameters | Time | Non-AMS (24) | Mild AMS (8) | Moderate/Severe AMS (8) | P value (A) | P value (B) | P value (C) |
|---|---|---|---|---|---|---|---|
| RBC ($10^{12}$/L) | Plain | 4.9 ± 0.3 | 5.0 ± 0.2 | 5.1 ± 0.4 | >0.05 | / | / |
| | 3 day | 5.5 ± 0.4 | 5.5 ± 0.5 | 5.7 ± 0.4 | >0.05 | / | / |
| | 7 day | 5.5 ± 0.4 | 5.5 ± 0.3 | 5.9 ± 0.5 | 0.04 | 0.055 | 0.008 |
| | 30 day | 5.6 ± 0.4 | 5.6 ± 0.4 | 5.6 ± 0.3 | >0.05 | / | / |
| Hb (g/L) | Plain | 151.5 ± 7.9 | 154.9 ± 3.8 | 156.8 ± 9.4 | >0.05 | / | / |
| | 3 day | 171.0 ± 11.0 | 171.3 ± 12.8 | 177.6 ± 12.4 | >0.05 | / | / |
| | 7 day | 170.3 ± 10.4 | 172.9 ± 4.6 | 184.8 ± 13.3 | 0.02 | 0.026 | 0.001 |
| | 30 day | 175.3 ± 9.3 | 174.0 ± 10.9 | 174.9 ± 12.7 | >0.05 | / | / |
| HCT(%) | Plain | 45.1 ± 2.6 | 45.9 ± 1.6 | 46.8 ± 3.0 | >0.05 | / | / |
| | 3 day | 52.4 ± 3.0 | 52.7 ± 3.7 | 54.6 ± 3.5 | >0.05 | / | / |
| | 7 day | 53.4 ± 6.1 | 56.1 ± 2.2 | 58.9 ± 4.7 | 0.03 | 0.285 | 0.015 |
| | 30 day | 50.3 ± 2.5 | 50.4 ± 3.4 | 50.2 ± 3.8 | >0.05 | / | / |
| SpO$_2$ (%) | Plain | 96.7 ± 0.7 | 96.4 ± 0.5 | 95.5 ± 1.8 | 0.014 | 0.078 | 0.003 |
| | 3 day | 88.3 ± 3.5 | 85.9 ± 3.7 | 86.8 ± 2.4 | >0.05 | / | / |
| | 30 day | 90.3 ± 2.7 | 90.8 ± 1.7 | 89.8 ± 2.7 | >0.05 | / | / |

**Note:**

Inter-group comparisons were made for the changes in blood parameters at different time points after rapid ascent to high altitude. The quantitative data followed a normal distribution and are presented as mean ± SD. P value(A) denotes the P-value of the ANOVA test, P value (B) represents the P-values for pairwise comparisons between the moderate/severe-AMS group and the mild-AMS group, P value (C) signifies the P-values for pairwise comparisons between the moderate/severe AMS group and the non-AMS group. The symbol "/" indicates that no statistically significant differences were found in the analysis of variance, thus pairwise comparisons were not conducted. AMS, acute mountain sickness.

and 52.7%. Individuals with moderate/severe AMS exhibited mean values of $5.7 \times 10^{12}$/L, 177.6 g/L, and 54.6%, respectively. Despite the slightly higher values observed in the moderate/severe AMS group compared to the mild AMS and non-AMS groups, these differences did not reach statistical significance ($P > 0.05$; Table 3).

On the seventh day following a rapid ascent to high altitude, individuals who did not experience AMS exhibited mean RBC, Hb, and HCT values of $5.5 \times 10^{12}$/L, 170.3 g/L, and 53.4%, respectively. In contrast, individuals with mild AMS had mean values of $5.5 \times 10^{12}$/L, 172.9 g/L, and 56.1%. Those with moderate/severe AMS had mean values of $5.9 \times 10^{12}$/L, 184.8 g/L, and 58.9%, respectively. The Hb levels of the moderate/severe AMS group were significantly higher than those of the mild AMS and non-AMS groups ($P < 0.05$; Table 3).

After 30 days at high altitude, there were no significant differences in the mean values of RBC, Hb, and HCT among the three groups ($P > 0.05$). Specifically, at this time point, individuals without AMS had mean RBC, Hb, and HCT values of $5.6 \times 10^{12}$/L, 175.3 g/L, and 50.3%, respectively. Those with mild-AMS exhibited mean values of $5.6 \times 10^{12}$/L, 174.0 g/L, and 50.4%, while individuals with moderate/severe-AMS had mean values of $5.6 \times 10^{12}$/L, 174.9 g/L, and 50.2%, respectively. The mean values of these parameters among the three groups showed no significant differences ($P > 0.05$; Table 3).

## The trends and differences in mean $SpO_2$ changes among the three groups following rapid ascent to high altitude

Within the initial 3 days following ascent to high altitude, all three cohorts experienced a substantial reduction in mean $SpO_2$ levels in comparison to their baseline measurements. The non-AMS, mild-AMS, and moderate/severe-AMS groups displayed average decreases of 8.3%, 10.5%, and 8.8%, respectively ($P < 0.001$). Between the third and thirtieth day post-rapid ascent, the mean $SpO_2$ levels of all groups began to show signs of improvement. Specifically, in contrast to the 3-day assessment, individuals without AMS demonstrated a mean increase of 2.0% in $SpO_2$, while those with mild-AMS and moderate/severe-AMS exhibited mean increases of 4.9% and 3%, respectively ($P < 0.05$; Table 2 and Fig. 4).

At sea level, the mean $SpO_2$ values for individuals without AMS, with mild AMS, and with moderate/severe AMS were 96.7%, 96.4%, and 95.5%, respectively. The mean $SpO_2$ value of the moderate/severe AMS group was found to be significantly higher than that of the non-AMS group ($P < 0.05$). However, on the third and thirtieth days after rapid ascent, there were no significant differences in the mean $SpO_2$ values among the three groups ($P > 0.05$; Table 3).

## Correlation between RBC, Hb, HCT, $SpO_2$ and the severity of AMS

The RBC count, Hb concentration, and HCT demonstrated a positive correlation with the severity of AMS at 7 days following rapid ascent; however, no such correlation was evident at other time points ($P < 0.05$; Figs. 5A–5C). Conversely, $SpO_2$ measured at sea level exhibited an inverse correlation with AMS, although this relationship was not significant after reaching the plateau ($P < 0.05$; Fig. 5D).

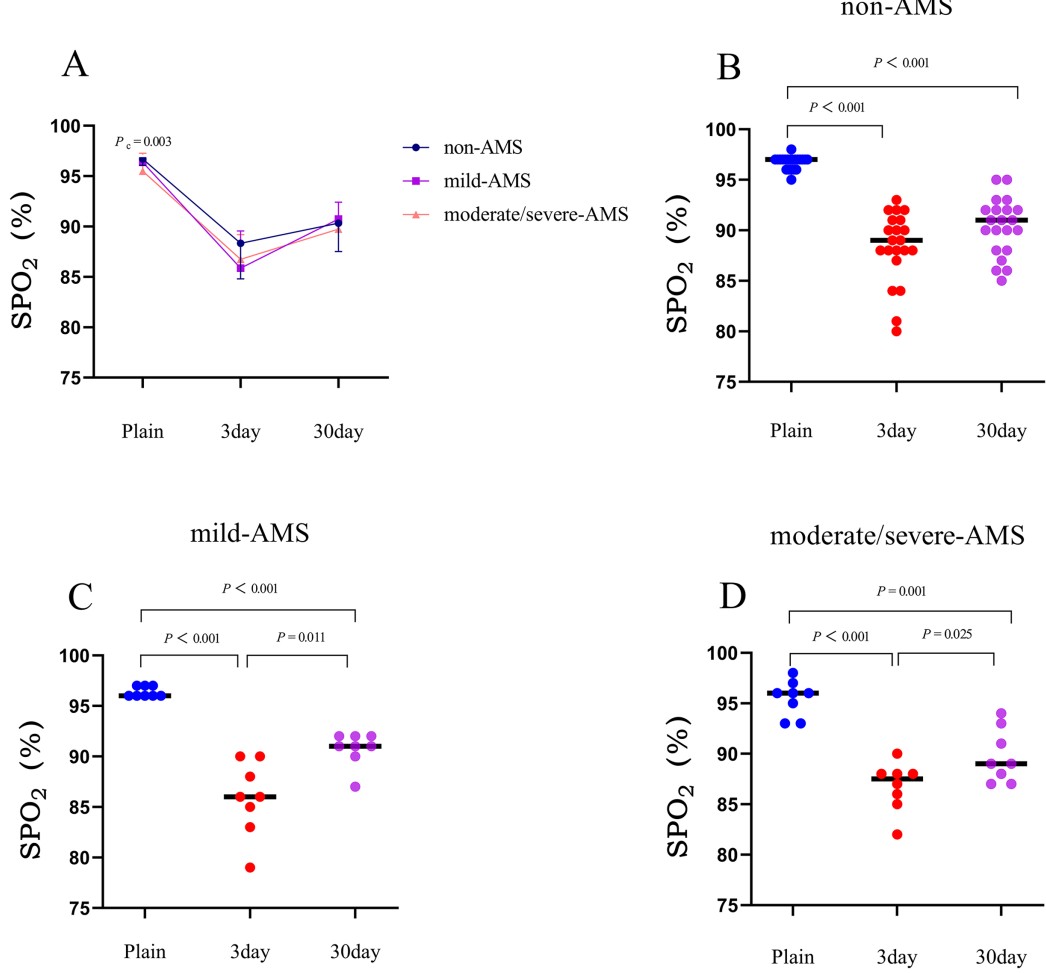

**Figure 4 The trends in mean changes of SpO₂ following rapid ascent to high altitude.** (A) The trends of SpO₂ variation in different groups before and after entering plateau. The difference of SpO₂ at 3 days and 30 days after entering plateau in non-AMS (B), mild-AMS (C), and moderate/severe-AMS (D).

## DISCUSSION

Prior studies have shown that the prevalence of AMS varies significantly among individuals and tends to increase at higher altitudes. The occurrence of AMS is influenced by a variety of factors, both intrinsic and extrinsic. Intrinsic factors encompass age (*Wu et al., 2018*), cardiopulmonary function (*Swenson, 2013*), BMI (*Wu et al., 2015*), physiological indicators (*Liu et al., 2023*), and genetic factors (*Mallet et al., 2023*), while extrinsic factors include altitude, mode of ascent, and season (*Muza, 2018*; *Talks et al., 2022*). While this study also noted an increase in the incidence of AMS with advancing age, the limited sample size precluded confident extrapolation of the findings. In the present research, surveys and blood samples were obtained from participants who experienced rapid ascent to high altitudes. A total of 40 valid questionnaires were gathered, revealing an incidence of AMS among participants at 40%, lower than the reported rate of 53% in

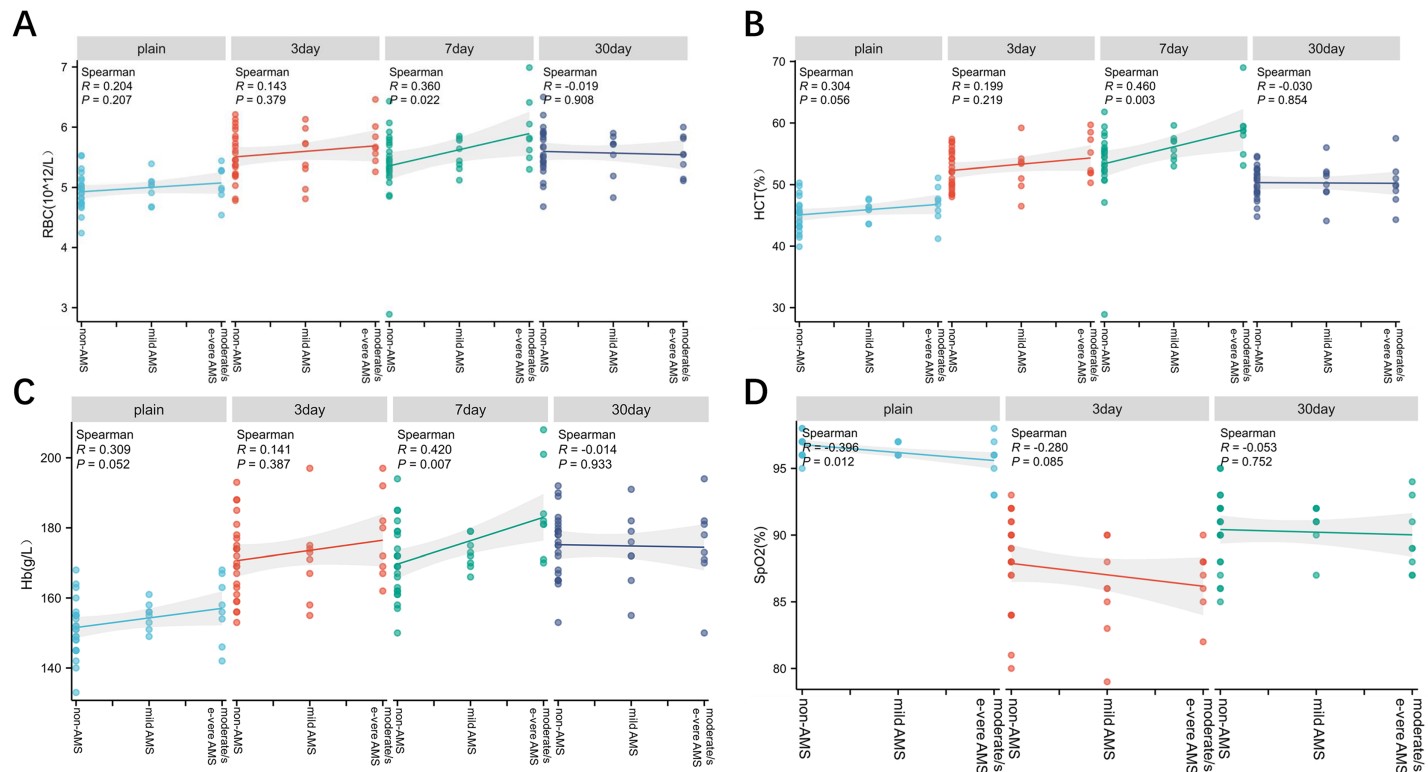

**Figure 5 Correlation between RBC, Hb, HCT, SpO₂ and the severity of AMS.** The Spearman was utilized to analyze the correlation between RBC, Hb, HCT, SpO₂ and the severity of AMS. The RBC (A), Hb (B), and HCT (C) showed a positive correlation with the severity of AMS at 7 days following rapid ascent, while SpO₂ (D) measured at sea level exhibited an inverse correlation with the severity of AMS.

existing literature (*Hackett, Rennie & Levine, 1976*). This variance may be linked to the comparatively lower altitude of 3,300 m in this study.

Following rapid ascent to high altitude, individuals often experience significant increases in parameters such as RBC, Hb, and HCT due to hypoxic stimulation (*Baart et al., 2021*; *Guo et al., 2022b*). However, our study observed distinct variations in the patterns of hematological parameter changes among individuals with moderate/severe AMS, mild AMS, and those without AMS. Specifically, individuals with moderate/severe AMS demonstrated a more rapid increase in RBC, Hb, and HCT levels on the third day post-ascent compared to those with mild AMS and non-AMS individuals, with peak mean values reached on the seventh day post-ascent. Furthermore, it is crucial to acknowledge that individuals experiencing moderate to severe AMS were administered oxygen therapy following rapid ascent to high altitudes in order to mitigate symptoms of hypoxia. While the duration of oxygen therapy was gradually reduced after 3 days of rapid ascent, the potential for notable effects on hematological parameters cannot be definitively dismissed.

The body's adaptation to hypoxia is mainly alleviated by regulating ventilation function and hypoxia signaling molecule HIF, which increases the proliferation ability of erythroid progenitor cells in bone marrow, promotes erythrocyte and angiogenesis, and increases tissue oxygen supply by increasing the transcription of downstream target genes EPO and

VEGF (*Li, Zhang & Zhang, 2018*). Volunteers with moderate/severe AMS suffered from a severe degree of hypoxia after a rapid approach to the plateau, resulting in a sustained rise in hemoglobin until day 7. After the acute hypoxia resolves, the hemoglobin value gradually decreases until 30 days, with no significant difference compared with the non-AMS group and mild AMS. In addition, the hydration status of the body may also have a greater impact on the concentration of hemoglobin due to the increase in water loss induced by hyperventilation after a rapid plateau.

According to prior research (*Li et al., 2018*), individuals experiencing more severe AMS exhibit larger increases in RBC, Hb, and HCT levels after 3 days at high altitude, attributed to variations in their adaptation to hypoxic conditions. This elevation is intended to improve oxygen delivery and surpasses that observed in individuals with milder AMS. Peak values are attained on the seventh day, after which RBC, Hb, and HCT levels begin to decline. Ultimately, by the thirtieth day at high altitude, there is no significant disparity in these values among individuals with varying degrees of AMS severity. However, the correlation analysis results indicated that RBC, HCT, and Hb exhibited a positive correlation with the severity of AMS after a 7-day acclimatization period at high altitude. By this juncture, individuals across various levels of AMS severity have largely acclimated to the hypoxic conditions at high altitudes. Following the seventh day of exposure to high altitude, HCT levels begin to decline in all three AMS groups, with a notably pronounced decrease observed in the moderate-severe AMS cohort. Furthermore, current studies indicate that red blood cell count or hemoglobin concentration in the bloodstream serves as the primary determinant of blood viscosity (*Mehri, Mavriplis & Fenech, 2018*). Elevated blood viscosity may elevate resistance to blood flow, decelerate blood flow rate, and lead to diminished oxygen binding, thereby exacerbating hypoxia (*Anderson et al., 2021*). HCT changes are closely related to changes in red blood cell count and blood volume, and a decrease in circulating blood volume is common in the early stages of a rapid plateau. Although hematological reactions at high altitudes have been studied for many years, they are still not fully understood. The right HCT is essential for the supply of oxygen (*Sitina, Stark & Schuster, 2024*). We believe that changes in HCT in the early stages of the rapid plateau may be the result of the body's regulation of its own oxygen supply. After 7 days of rapid entry into the plateau, the body gradually adapts to the hypoxic environment, and the higher HCT increases the blood viscosity, which is not conducive to oxygen supply, and the HCT begins to decline to reach a new equilibrium.

Furthermore, the process of red cell pheresis has been shown to be beneficial in improving the oxygen-deficient state of patients with polycythemia (*Liu et al., 2013*). These results suggest that a decrease in hematocrit levels beyond the seventh day at high altitudes may have a positive impact on tissue oxygenation. Moreover, our study revealed that there were no significant disparities in the alterations of red blood cell count, hemoglobin levels, and hematocrit mean values between individuals experiencing mild AMS and those without AMS. Additionally, the patterns of change in these parameters following rapid ascent were similar, possibly due to the lesser severity of hypoxia and quicker compensatory responses in these individuals.

Another important indicator reflecting the body's adaptation to hypoxia environment is $SpO_2$. Following rapid ascent to high altitude, individuals experiencing AMS generally have lower $SpO_2$ levels compared to those without AMS, indicating more severe tissue and organ hypoxia (*Dunnwald et al., 2021*; *West, 2014*). Our study also showed that $SpO_2$ levels were lower in the mild-AMS and moderate/severe-AMS groups, indicating more severe hypoxia compared to the non-AMS. Additionally, we found that individuals in the moderate/severe-AMS group had lower $SpO_2$ levels at baseline compared to those without AMS, suggesting that baseline $SpO_2$ levels may influence the occurrence of AMS. Studies have shown that oxygen saturation $SpO_2$ has a good predictive effect on AMS (*Goves et al., 2024*). Our study also found that the oxygen saturation of volunteers with moderate/severe AMS was lower at sea level and slightly lower than that of volunteers without AMS after a sharp advance to the plateau, and there was a positive correlation between oxygen saturation and AMS severity, and sea level oxygen saturation may be an important factor in predicting the occurrence of more severe AMS. Monitoring of oxygen saturation during ascent may be of higher value for the prediction of severe AMS and altitude adaptation. BMI is also one of the influencing factors of AMS, obesity may increase the incidence of AMS (*Guo et al., 2022a*), in order to avoid the influence of this factor on the experiment, the BMI of the volunteer population was less than 25, and there was no significant difference between groups at the time of enrollment.

Previous studies have consistently demonstrated a correlation between the onset of AMS and decreased $SpO_2$ levels (*Dunnwald et al., 2021*; *Wei et al., 2022*), indicating that individuals affected by AMS typically exhibit notably lower $SpO_2$ levels compared to those unaffected by AMS. This association remains significant even after adjusting for other variables (*Harrison et al., 2013*). $SpO_2$ has important clinical value in the prediction, occurrence, development, and treatment of AMS (*Cobb et al., 2021*; *Guo et al., 2022a*). The potential reasons could be that in populations with low blood oxygen saturation, the sensitivity of the body to hypoxia increases. As altitude increases, the concentration of oxygen in the air gradually decreases. This means that in high-altitude areas, even a slight decrease in blood oxygen saturation may lead to symptoms of hypoxia in the body. Therefore, we believe that baseline $SpO_2$ may be a potential indicator reflecting the occurrence of moderate-severe AMS.

In fact, there are several participant's basic characteristics may also affect AMS. For example, the impact of gender on AMS has been a subject of interest in recent research. According to a meta-analysis, there is evidence to suggest that there are sex-based differences in the prevalence of AMS. The study found that women have a statistically significant higher prevalence rate of AMS compared to men, with a relative risk (RR) of 1.24, regardless of age or race (*Hou et al., 2019*). This indicates that women may be more susceptible to developing AMS when ascending to high altitudes. The potential mechanisms behind this difference are not fully understood, but some hypotheses have been proposed. One hypothesis suggests that hormonal differences between men and women may play a role. Estrogen, present in higher levels in women, is thought to upregulate vascular endothelial growth factor (VEGF) expression, which can increase vascular leakage and contribute to intracranial hypertension, a factor in AMS

(*Schoch, Fischer & Marti, 2002*). Another mechanism relates to the concentration of erythropoietin (EPO), which is influenced by testosterone, an androgen. Men may have an advantage at high altitudes due to the promotion of erythropoiesis by testosterone, which could improve oxygen carrying capacity and reduce the prevalence of AMS (*Ding et al., 2014*). Therefore, further study should comprehensively analyze the differences between different genders, age and fitness status on the basis of increasing the sample size, so as to provide ideas for the pathological mechanisms of AMS and potential gender-specific prevention strategies.

Nevertheless, the present study is subject to certain limitations. Specifically, this study was limited by the fact that there were only eight volunteers with moderate/severe AMS; Hypoxia-related markers such as EPO, VEGF, and HIF were not detected. Blood samples were not obtained on the initial and final days following rapid ascent, and certain stress indicators were omitted. Additionally, since the personnel entering the plateau for training are young males of Han descent with perennial sports training background, this may lead to different degrees of bias in the data analysis results of the present study. Therefore, future research endeavors should carefully account for fundamental demographic characteristics (including age, gender, ethnicity, height, weight, *etc.*) of the study cohort to mitigate potential sources of bias in the research outcomes.

## CONCLUSION

The current data suggests that individuals experiencing moderate to severe AMS predominantly modulate tissue oxygenation during the initial phases of rapid ascent to high altitudes by promptly elevating RBC and Hb levels to acclimatize to hypoxic conditions. As the ascent progresses, the hypoxic environment ameliorates, resulting in negligible variations in RBC, Hb, and other parameters among individuals with varying degrees of AMS. Additionally, our findings propose that individuals with lower baseline $SpO_2$ levels may be at a heightened risk of developing moderate to severe AMS.

### Funding

This work was supported by the National Natural Science Foundation of China (grant number: 81971778) and the Youth Talent Project Plan of Air Force Medical Center (grant number: 22BJQN009). The funders had no role in study design, data collection and analysis, decision to publish, or preparation of the manuscript.

### Grant Disclosures

The following grant information was disclosed by the authors:
National Natural Science Foundation of China: 81971778.
Youth Talent Project Plan of Air Force Medical Center: 22BJQN009.

### Competing Interests

The authors declare that they have no competing interests.

## Author Contributions

- Zhicai Li conceived and designed the experiments, performed the experiments, analyzed the data, prepared figures and/or tables, authored or reviewed drafts of the article, and approved the final draft.
- Jun Xiao conceived and designed the experiments, performed the experiments, analyzed the data, prepared figures and/or tables, authored or reviewed drafts of the article, and approved the final draft.
- Cuiying Li conceived and designed the experiments, authored or reviewed drafts of the article, and approved the final draft.
- Xiaowei Li performed the experiments, authored or reviewed drafts of the article, and approved the final draft.
- Daoju Ren performed the experiments, authored or reviewed drafts of the article, and approved the final draft.

## Ethics

The following information was supplied relating to ethical approvals (*i.e.*, approving body and any reference numbers):

The studies involving humans were approved by the Institutional Review Board of the Air Force Medical Center (NO. 2022-05-YJ07).

## Data Availability

The datasets used and/or analysed during the current study available at Zenodo: Li, C. (2024). The raw data set for people entering plateau. [Data set]. Zenodo. https://doi.org/10.5281/zenodo.14093574.

The raw data are available in the Supplemental File.

## Supplemental Information

Supplemental information for this article can be found online at http://dx.doi.org/10.7717/peerj.18738#supplemental-information.

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
