# Peer review of "Correlation between hematological indicators in acclimatized high-altitude individuals and acute mountain sickness"

_PeerJ, doi:10.7717/peerj.18738_

## Round 0.1 · original submission · Major Revisions

Dear Dr. Li, I ask you to carefully respond to each of the reviewers' comments and improve the manuscript. It is especially important to improve the manuscript on the reviewers' fundamental comments. Your article is of great practical importance, and it will probably arouse great interest among readers from different regions of the world. Try to make sure that the publication of this article is successful.

Reviewer 1 ·

Basic reporting

Lines: 54, 55 and 56, mention that the changes in exposure to altitude occur from 3000 meters onwards, verify this information because these occur from 2500 meters onwards as indicated by most of the literature.
Line 72: Population, they do not indicate how many individuals were men and how many women, as we know the response to exposure to hypoxia is different in men and women.
The results mention numerical data that can also be seen in the tables and graphs. In order not to repeat the information twice, they should only show numerical data in the tables and graphs and not in the results section.
The study does not have the discussion that would allow us to better understand the changes that have occurred over time and that would also allow us to better understand each of the phenomena, having to give a possible explanation for these changes.
The figures should allow us to better understand the differences in the changes of the subjects with and without AMS according to the type or intensity; by doing them separately, the difference in the behavior of these variables cannot be detected.
They should improve the format of the work, complete the discussion and use either figures or tables but not show the same information repeatedly.

Experimental design

Well defined research questions, but there are some aspects like basic. definitions as high altitude, age and sex should be well defined

Validity of the findings

Once changes are made, the results could be valid and applicable

Additional comments

none

·

Basic reporting

Respected author,

I appreciate the opportunity to review and share my opinions on the article titled “Correlation between hematological indicators in acclimatized high-altitude individuals and acute mountain sickness” (#2024:06:101938:0:3:REVIEW). I believe this work addresses a topic of great relevance, given the increasing population exposed to high-altitude hypoxia. Below are my comments

Introduction

• Although the introduction establishes a solid foundation regarding acute mountain sickness (AMS), it would be beneficial to include a more thorough analysis of individual variability in altitude response, as well as acclimatization, genetic, and training factors. A more comprehensive review of these elements could better con

• While previous studies are cited, the introduction does not develop a robust theoretical framework that examines in greater depth the physiological mechanisms behind the hematological differences observed in patients with AMS. It would be pertinent to conduct a more exhaustive review of the existing literature to provide a stronger theoretical foundation for the proposed hypothesis.

• It is important to clearly define the terms "high altitude" and "very high altitude," as their interchangeable use may cause confusion. A precise definition from the outset of the document would facilitate understanding and following the content.

• While the need to investigate hematological parameters is mentioned, it would be valuable to include a summary of previous studies that have explored this relationship, beyond the statement that a gap in the literature persists regarding specific research on hematological changes in individuals experiencing different degrees of AMS.

Experimental design

Methodology

• The environmental context in which the samples were taken is not explicitly mentioned. Factors such as humidity, temperature, exact altitude, and barometric pressure are critical variables in this type of study, as any fluctuation in these conditions can affect physiological parameters.

• The usual environment of the subjects prior to their exposure to altitude hypoxia is not described. At what altitude above sea level do they normally reside? What is the average humidity and temperature in those areas? This information is crucial, as the subjects' previous physiological adaptations to their environment could influence their response to hypoxia.

• Regarding exclusion criteria, these should be more explicit and consider aspects that could influence hematological changes, such as sports level or history and the presence of kidney or bone diseases that interfere with erythropoiesis. Furthermore, it is unclear whether the study included both men and women (although supplementary data do indicate measurements in both genders). If so, it is essential to detail in which phase of the menstrual cycle the measurements in women were taken, as this factor can impact the results, and a clear sex discrimination should be included in the analyses.

• The absence of the ethical committee registration number and the clinical study protocol is concerning, as ensuring transparency and compliance with ethical regulations is fundamental.

• Regarding the scale used, while it is mentioned that it was based on the Lake Louise Acute Mountain Sickness Score, it is indicated that it was modified. It would be advisable to present the validation study of this modified scale to support its use in the specific context of the study and within this population.

• The manuscript does not mention whether factors that could influence the results, such as physical activity, diet, hydration status, and substance consumption (caffeine, alcohol, medications) during the stay in the hypobaric hypoxia environment, were controlled. It should be noted that plasma volume variation due to fluid intake interferes with hematological results.

• While methods to measure hemodynamic and hematological parameters are mentioned, there is insufficient information regarding methodological consistency. For example, were measurements taken at the same time each day, under standardized conditions for all subjects? How were the EDTA tubes with blood samples stored? How much time elapsed after collection before each sample was analyzed? What equipment and reagents were used for blood sample analysis and SpO₂ measurement? The lack of standardization could introduce biases in the results.

• The hematological analysis process lacks detailed methodology. For example, details on how the samples were processed and what confounding factors were considered during analysis are omitted. This information is essential to ensure the reproducibility of the study.

• It is important to specify whether the study included both men and women. If so, it should indicate in which phase of the menstrual cycle measurements were taken in women, as this can significantly influence results. Additionally, the results should be analyzed by sex.

• Since the response to hypoxia can vary considerably based on sex, age, and prior fitness status, it would be pertinent to conduct subgroup analyses to identify whether certain demographic groups respond differently. The absence of these analyses may obscure important findings or lead to inaccurate conclusions.

Validity of the findings

Results

• The organization of the results facilitates comparison between the different severity groups of AMS, allowing for a clear identification of patterns in hematological changes and oxygen saturation (SpO₂).

• Although the results indicate that individuals with severe/moderate AMS show greater increases in hematological parameters, significant differences were not observed at certain times. This suggests that physiological responses could be more complex than anticipated, perhaps influenced by unmeasured factors such as individual acclimatization, hydration status, or genetic factors, which should be strengthened in the discussion.

• While the article indicates the presence of relationships through statistical significance (e.g., p-values), it does not explicitly present correlation coefficients (such as Pearson or Spearman correlations) in the results section, despite the title of the manuscript being “Correlation between hematological indicators in acclimatized high-altitude individuals and acute mountain sickness.”

Discussion

• The discussion tends to generalize the results without adequately considering the study's limitations, particularly regarding sample size and participant characteristics. A more nuanced interpretation that takes these limitations into account would strengthen the scientific rigor of the study.

• Although previous research is mentioned, the possible physiological explanations justifying why individuals with moderate/severe AMS showed a more significant increase in hemoglobin (Hb) and hematocrit (HCT) levels compared to other groups at certain times are not explored in depth. A deeper analysis of the underlying physiological mechanisms would enrich the interpretation of these findings.

• While changes in SpO₂ are mentioned, the discussion does not delve into the relevance of these findings or how they may influence altitude adaptation or the onset of AMS in individuals with varying severity levels. There is minimal exploration of the clinical implications of the results. The discussion should address how these findings can inform practical strategies for altitude acclimatization and AMS management.

• The discussion lacks a thorough analysis of how protein systems such as HIF-1 (hypoxia-inducible factor 1) and erythropoietin (EPO) intervene in physiological responses to different levels of AMS. These proteins play a key role in hypoxia adaptation, and their influence may vary depending on the severity of AMS. A more detailed analysis of these mechanisms would significantly enrich the study's discussion.

Additional comments

The reviewed article addresses hematological responses and oxygen saturation in individuals rapidly ascending to high altitudes (in a descriptive manner), which entails the risk of developing AMS. Although the research holds significant value and remains relevant for those studying the effects of altitude on human physiology, the topic is not entirely novel. The effects of AMS and hematological adaptations to altitude have been the subject of multiple previous studies integrating the study of hematological, ventilatory, and metabolic variables with cellular response mechanisms based on protein synthesis and inflammatory markers.

Although the article enriches our understanding of AMS and hematological responses, it could benefit from more innovative contributions to enhance its scientific novelty. It could benefit from greater diversity in the sample and a deeper exploration of other physiological systems that interact during altitude acclimatization. The article addresses a crucial area of research regarding hematological changes during rapid ascents to high altitudes and their relationship with AMS. However, several weaknesses in the methodology, results, and discussion sections undermine its overall impact. Strengthening these areas would allow the authors to improve the scientific rigor and relevance of their findings significantly.

Reviewer 3 ·

Basic reporting

- Language is clear and concise throughout
- It is noted that "specific relationship between AMS and alterations in blood parameters remains unclear", however there is relevant literature not referenced here: https://bjsm.bmj.com/content/57/14/906.abstract & https://www.sciencedirect.com/science/article/pii/S1569904824001113 for lung function parameters.
- Professional article structure, figures, tables. Raw data properly shared.
- Self contained with relevant results to hypotheses but this part could be strengthened with appropriate data analysis methods and more consistent reporting as described in the sequel.

Experimental design

- Within aims and scope.
- Research question well-defined.
- Rigorous investigation appears to have been performed.
- Methods are probably inappropriately described in the statistical analysis part and further results are at stake as follows: It is reported that the t-test and ANOVA were used for data analysis, however it is not clear whether a repeated measures ANOVA was used (time is the repeated factor term), while AMS groups appear to have been treated/analysed separately. Current description does not take inherent correlations into account. Not much is stated regarding comparisons between AMS groups (I have verified that HCT does not differ between groups at the different time points). Furthermore, the relevance of the t-test is questionable in this design. Not clear if gender differences were considered within the model used (as an extra covariate).

Validity of the findings

- Given that studies with null findings can be considered in the journal, specifically reporting on AMS group differences and other baseline information importance (e.g. sex, BMI) could be considered.
- Data have been provided in Excel format. P-values reported are at stake given that the use of repeated measures ANOVA (or a general linear mixed model) is not clear.
- Reporting on AMS prevalence in the main text and conclusions is overstated while it is not mentioned in the title and abstract.

---

## Round 0.2 · Minor Revisions

Dear authors, I kindly ask you to carefully improve the manuscript in accordance with the reviewers' comments and send the final version of the manuscript for acceptance for publication.

·

Basic reporting

The modifications in the introduction allow for a broader and more coherent view of the research topic.

Experimental design

The modifications to the methodology enhance the understanding of the experimental design and provide a solid foundation for future replication of the study

Validity of the findings

Appropriate and well-aligned with the intended objectives of the article.

Reviewer 3 ·

Basic reporting

Editing for English is needed, parts of the text are hard to follow.
Sufficient field background is now provided.
Figures are of low quality, could be better. Raw data are given.
Results relevant to initial hypotheses.

Experimental design

Within aims and scope.
It is stated how research fills an identified knowledge gap.
Methods are described with sufficient detail. Replication is possible, I have not tried to replicate results.

Validity of the findings

All underlying data have been provided.
Conclusions are well stated, linked to original research question. Editing for language is needed.

Additional comments

The authors have edited the text according to suggestions, clarifying methods followed. Figure quality is poor.

---

## Round 0.3 · accepted · Accept

Dear authors, I am pleased to inform you that this article has been accepted for publication. I hope that you will continue to research this topic.